# Platelet rich Plasma in Achilles Tendon Healing 2 (PATH-2) trial: protocol for a multicentre, participant and assessor-blinded, parallel-group randomised clinical trial comparing platelet-rich plasma (PRP) injection versus placebo injection for Achilles tendon rupture

Joseph Alsousou,[1] David J Keene,[2] Philippa A Hulley,[3] Paul Harrison,[4] Susan Wagland,[2] Christopher Byrne,[5] Michael Maia Schlüssel,[6] Susan J Dutton,[6] Sarah E Lamb,[2,6] Keith Willett[2]

JA and DJK contributed equally.

For numbered affiliations see end of article.

**Correspondence to**
Joseph Alsousou;
josephalsousou@doctors.org.uk

## ABSTRACT

**Background** Achilles tendon injuries give rise to substantial long-lasting morbidity and pose considerable challenges for clinicians and patients during the lengthy healing period. Current treatment strategies struggle to curb the burden of this injury on health systems and society due to lengthy rehabilitation, work absence and reinjury risk. Platelet-rich plasma (PRP) is an autologous preparation that has been shown to improve the mechanobiological properties of tendons in laboratory and animal studies. The use of PRP in musculoskeletal injuries is on the increase despite the lack of adequately powered clinical studies.

**Methods and design** This is a multicentre randomised controlled trial to evaluate the efficacy and mechanism of PRP in patients with acute Achilles tendon rupture (ATR). All adults with acute ATR presenting within 12 days of the injury who are to be treated non-operatively are eligible. A total of 230 consenting patients will be randomly allocated via a remote web-based service to receive PRP injection or placebo injection to the site of the injury. All participants will be blinded to the intervention and will receive standardised rehabilitation to reduce efficacy interference. Participants will be followed up with blinded assessments of muscle–tendon function, quality of life, pain and overall patient's functional goals at 4, 7, 13, 24 weeks and 24 months post-treatment. The primary outcome is the heel-rise endurance test (HRET), which will be supervised by a blinded assessor at 24 weeks. A subgroup of 16 participants in one centre will have needle biopsy under ultrasound guidance at 6 weeks. Blood and PRP will be analysed for cell count, platelet activation and growth factor concentrations.

**Ethics and dissemination** The protocol has been approved by the Oxfordshire Research Ethics Committee (Oxfordshire Research Ethics Committee A, reference no 14/SC/1333). The trial will be reported in accordance with the CONSORT statement and published in peer-reviewed scientific journals.

**Trial registration number** ISRCTN: 54992179, assigned 12 January 2015. ClinicalTrials.gov: NCT02302664, received 18 November 2014. UK Clinical Research Network Study Portfolio Database: ID 17850.

## INTRODUCTION

Achilles tendon rupture (ATR) incidence is 21/100 000/year.[1] ATR accounts for 20% of all tendon ruptures and leads to major health-care and societal costs. The current treatment strategies are either augmentation with surgical suture or immobilisation in a cast or boot. The mechanical and biological properties of healed tendons appear never to match those of the original intact tendons, leading to high risk of further injury (5%–15%) or reduced function.[2–4] Moreover, a Cochrane review reported mean ATR rehabilitation and work absence of 63–108 days.[5]

Platelet-rich plasma (PRP) is a derivative of the patient's own blood (autologous blood) that contains a supraphysiological concentration of platelets. Platelets contribute to injury healing by releasing an ordered sequence of growth factors, cytokines and an array of bioactive proteins in soluble and membrane-bound forms over the lifespan of the platelets.[6 7] These include transforming growth factor (TGF-β1 and TGF-β2), platelet-derived growth factor (PDGF-AA, PDGF-AB and PDGF-BB), vascular endothelial growth factor (VEGF-A and VEGF-C), insulin-like growth factor 1 (IGF-1) and epidermal growth factor.[7 8] These factors recruit a range of cell

types, including the injured tendon's tenocytes, leucocytes and local stem cells, and promote the healing pathway. PRP enhances angiogenesis, stem cell homing, local cell migration, proliferation and differentiation coupled with the deposition of proteins such as collagen which plays a key role in enabling the restoration of normal tissue structure and functional proliferation of human tenocytes.[8 9]

PRP is prepared from autologous blood using gravitational platelet sequestration centrifugation, cell separators or selective filtration technology (plateletpheresis). Each preparation technique has been evidenced to result in significant differences in yields, concentration, purity, viability and activation status of the platelets.[10] Each of these variables will not only influence the eventual concentrations of the bioactive proteins but may also affect the clinical efficacy of each PRP preparation.[11] We selected a centrifugation technique with highly standardised preparation protocol that offers consistently viable and active PRP with a high concentration of platelets and leucocytes (leucocyte-rich PRP (L-PRP)).[12]

To date, there is only one randomised controlled trial (RCT) that has assessed PRP in ATR, and in this study of 30 participants treated surgically, no effect of platelets on radioisometrical tendon contraction was seen[13]; the use of PRP as an adjunct to open surgical repair may have obscured any effect of PRP on healing. In a case–control study of 12 athletes treated with PRP, positive effects at 32 months after treatment were demonstrated.[14] Less tendon thickening and higher concentrations of TGF-β and other growth factors were seen in the intervention patients, who also regained range of motion faster and returned to gentle running earlier.[14] Systematic reviews[15 16] concluded that there are encouraging signs that PRP could be developed as an effective tendon therapy, with potentially faster recovery and, possibly, a reduction in injury, with no adverse reactions described.[16] Both reviews emphasised the need for an adequately powered RCT to establish PRP efficacy with disease-specific outcome measures.

We describe the first multicentre RCT to evaluate the efficacy and mechanism of PRP in patients with acute ATR, where adequate power and robust, validated, objective and participant-reported outcome measures will ensure successful efficacy evaluation. Our aim is to investigate if the efficacy signal for PRP identified in basic science translates to improved mechanical muscle–tendon unit recovery in patients with acute ATR. The primary objective is to evaluate the clinical efficacy of PRP in acute ATR in terms of mechanical muscle–tendon function. The secondary objectives are to evaluate the clinical efficacy in terms of participant reported functional recovery, pain and quality of life; determine the key components of PRP that contribute to its mechanism of action; further understand in an immunohistochemical substudy the mechanisms of PRP which may account for its clinical efficacy and identify the histological pathways that PRP may alter to exert its effects.

## METHODS
### Study design
A prospective multicentre, participant and outcome assessor blinded randomised placebo-controlled superiority trial, which aims to evaluate the clinical efficacy of PRP in acute ATR in terms of recovery of the tendon function. Two hundred and thirty participants will be randomised to receive a PRP or placebo injection after attending the orthopaedic or trauma outpatient clinics within 12 days of the injury in at least 18 National Health Service (NHS) hospitals. Two substudies are embedded in the main study to evaluate the mechanism of action and PRP components. The substudies are: (1) immunohistochemistry analysis of ultrasound-guided needle

**Table 1** Participant's inclusion and exclusion criteria

| Inclusion criteria | Exclusion criteria |
|---|---|
| ► Willing and able to give informed consent for participation | ► Achilles tendon injuries at the insertion to the calcaneum or at the musculotendinous junction |
| ► Age ≥18 years | ► Previous major tendon or ankle injury or deformity to either lower leg |
| ► Diagnosed with an acute complete Achilles tendon rupture | ► History of diabetes mellitus |
| ► Presenting within and receiving study treatment within 12 days postinjury | ► Known platelet abnormality or haematological disorder |
| ► Patients in whom the decision has been made for non-operative treatment | ► Current use of systemic cortisone or a treatment dose of an anticoagulant |
| ► Ambulatory prior to injury without the use of walking aids or assistance of another person | ► Evidence of lower limb gangrene/ulcers or peripheral vascular disease |
| ► Able and willing to comply with all study requirements | ► History of hepatic or renal impairment or dialysis |
| ► Able to attend the 24-week follow-up at a PATH-2 study hospital site | ► Pregnant or breast feeding |
| | ► Currently receiving or has received radiation or chemotherapy within the last 3 months |
| | ► Has inadequate venous access for drawing blood |
| | ► Has any other significant disease or disorder which, in the opinion of the recruiting clinician, may either put the participant at risk because of participation in the study, influence the result of the study or influence the patient's ability to participate in the study |

PATH-2, Platelet Rich Plasma in Achilles Tendon Healing 2.

**Table 2** Primary outcome measures

| Primary outcome measure: HRET | |
|---|---|
| Measurement | Work (J) of each limb in heel-rise test |
| Analysis variable | LSI |
| Description | $LSI = \dfrac{\text{Injured Limb measurement}}{\text{Uninjured Limb measurement}} \times 100$ |
| Method of aggregation | Mean±SD |
| Time point | 24 weeks postintervention |
| HRET other variables | Number of heel raises performed by each limb |
| | Maximum displacement during the HRET for each limb (cm) |

HRET, heel-rise endurance test; LSI, Limb Symmetry Index.

biopsies from 16 participants at one centre (Oxford) and (2) whole blood and PRP component analysis in all participants. A whole blood sample will be obtained from each participant prior to intervention. In the PRP group, a small volume of PRP will be used to analyse the biological components. Blood and PRP analysis will be carried out at a central specialised laboratory (Institute of Inflammation and Ageing Laboratory at the University of Birmingham).

Through immunohistochemical, PRP and blood analysis, the potential mechanism of action will be studied to determine the key components of PRP that contribute to its effect. Linking the outcomes and the embedded laboratory analysis will allow us to evaluate the effect of variability of this biological product on the clinical outcome.

### Study participants

All patients with acute ATR attending outpatient trauma or orthopaedic clinic within 12 days of sustaining the injury are eligible for the trial if they meet all of the inclusion criteria and none of the exclusion criteria (table 1). At least 18 UK NHS hospitals will participate to recruit the required 230 patients. A list of participating sites to date can be found on the Platelet Rich Plasma in Achilles Tendon Healing 2 (PATH-2) website[17] and in an online supplementary table .

### Centre recruitment

A minimum of 18 NHS hospital orthopaedic trauma/outpatient clinics will recruit 230 participants for the trial. Each site will identify a surgeon to act as PATH-2 Principal Investigator (PI). The PI oversees the study protocol implementation at each site, uses links with local physiotherapy departments to facilitate the standardised rehabilitation and arranges for a blinded physiotherapist who will be the assessor for the heel-rise endurance test (HRET) primary outcome measurement. The trial team will assess each centre to ensure the site is equipped with appropriate resources to deliver the project and meet recruitment targets.

### Participant recruitment

Participants will be identified in the outpatient trauma or orthopaedic clinic. The treating surgeon or extended

scope physiotherapist will confirm the diagnosis, appropriateness for conservative treatment and study eligibility. A member of the research team at the site will carry out the informed consent process, baseline data collection and randomisation.

### Informed consent

The timeframe between attending clinic and receiving the intervention is relatively short. To help raise awareness of the study during the clinic wait, sites will be provided with written study participant information (including posters) to display in clinic where potential participants are waiting to be seen by the surgeon or extended scope physiotherapist.

The attending surgeon or extended scope physiotherapist will meet with the participant for the clinical examination and decide if the management will be operative or non-operative. If non-operative, participants will be informed of the study and given a patient information sheet (PIS). The participant will be allowed as much time as practically possible in this type of acute injury to consider the information and will have the opportunity to ask questions of the attending clinician and a member of the research team.

The person who obtains the consent must be a registered health or medical practitioner who is Good Clinical Practice (GCP) trained and has been authorised to do so by the PI. In most sites, this is a research nurse or physiotherapist or surgeon who will be a part of the local NHS Trust or the local National Institute of Health Research (NIHR) clinical research network. The participant must sign and date the informed consent form before any study-specific procedures are performed. Participants will also be asked to consent to use of their data, biological specimens and videos of their HRET test (view is of leg only). A copy of the signed informed consent form will be given to the participants, and one copy will be sent to the study coordinating team. The original signed consent form will be retained in the medical notes and a copy held in the Investigator Site File.

### Baseline assessment

Baseline data are collected immediately following confirmation of eligibility and consent. Background and

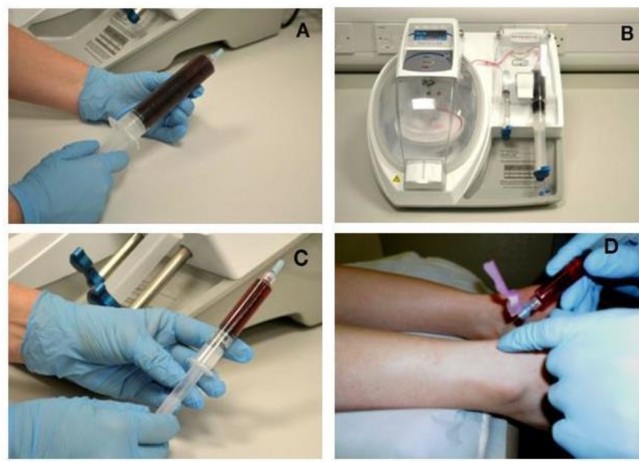

**Figure 1** Making autologous PRP in the PATH-2 study. (A) A whole blood sample is taken. (B) The Magellan Autologous Platelet Separator System is used to produce PRP. (C) The resulting PRP is collected in a syringe for injection into the Achilles tendon rupture gap. (D) The injection is delivered in the tendon rupture gap. PATH-2, Platelet Rich Plasma in Achilles Tendon Healing 2; PRP, platelet-rich plasma.

demographic information is collected including: general health, current medication, allergies, smoking, alcohol use, age, sex, employment status, type of employment, activities of daily living, sport and recreational activities prior to injury, the activity that led to the injury, previous history, height and weight. The Achilles Tendon Rupture Score (ATRS), a validated and responsive participant-reported outcome measure (PROM), is collected.[18 19] In addition, PROMs: Patient-Specific Functional Scale

**Table 3** Secondary outcome measures

| Time point | Outcome measures |
| --- | --- |
| Baseline | ATRS, PSFS, SF-12 (preinjury and postinjury), VAS (pretreatment) Substudy 1: blood sample (both groups), PRP analysis (PRP group) |
| 2 weeks | Daily record of post-treatment pain using daily pain diary (VAS) |
| 6 weeks | Substudy 2: tendon needle biopsy under ultrasound guidance analysis (16 participants, 8 in each arm, central site) Immunohistochemistry analysis |
| 4, 7 and 13 weeks | ATRS, PSFS and SF-12 Recorded by telephone call or during outpatient visit |
| 24 weeks | ATRS, PSFS, SF-12, HRET Conducted via assessment at outpatient visit |
| 24 months | ATRS, PSFS and SF-12 recorded by telephone call |

ATRS, Achilles Tendon Rupture Score; HRET, heel-rise endurance test; PRP, platelet-rich plasma; PSFS, Patient-Specific Functional Scale; SF-12, 12-Item Short Form Health Survey; VAS, Visual Analogue Scale.

(PSFS),[20 21] Visual Analogue Scale (VAS) pain indicator[22] and the acute version 12-Item Short Form Health Survey (SF-12) assessment.[23] The latter is collected as recalled before injury and also on the day of treatment (table 2). The recall of SF-12 health status and physical ability in the 4 weeks prior to the injury is a valid method considering the acute occurrence of the injury and the short period to recruitment (up to 12 days).

### Randomisation

Participants will be randomly allocated (1:1) to the two groups (treatment and placebo groups) via a central computer-based allocation randomisation system provided by the Oxford Clinical Trials Research Unit (OCTRU). The randomisation will use minimisation, stratified by study site and age group (<55 and ≥55 years) and with a probabilistic element of 0.8 to reduce predictability for this multicentre trial while retaining balance within centre and ensure overall balance at the end of the study.[24] Participants will remain blind to their allocated treatment throughout the study.

### Interventions

The two trial groups are:
► PRP injection (treatment group): After local anaesthetic injection, PRP is injected into the tendon rupture gap.
► Imitation injection (placebo group): After local anaesthetic injection, a needle is introduced into the tendon rupture gap, held briefly and withdrawn without injecting and without disturbing the biological haematoma.

Immediately after randomisation, up to 55 mL of venous blood is withdrawn from the participant regardless of the random allocation. The exact amount of blood and the volume of PRP injected are not stated in the PIS as this information has the potential to non-blinded participants.

PRP will be prepared from the venous blood by a study-specific centrifuge (MAG 200 MAGELLAN Autologous Platelet Separator, Arteriocyte Medical Systems) in or near the clinic (figure 1). This device has been found to produce around a fivefold concentration of platelets with 76% platelet recovery.[12] This is a fully automated system, requiring a sterile PRP disposable kit (MDK 300/MDK 300-1, Arteriocyte Medical Systems) and a single preparation step, reducing the risk of protocol deviation while in use.

Both interventions will be delivered by a clinician or extended scope physiotherapist while maintaining the participant's blinding to the allocation. Both interventions are delivered by the same technique. The patient is positioned lying face down on a treatment bed with the tendon exposed. The tendon gap is palpated clinically to identify the injection site. The area is cleaned, and 1–2 mL local anaesthetic is used to anaesthetise the skin only. In PRP injection group, half of the PRP is injected in the tendon gap (figure 1D), and the remaining PRP

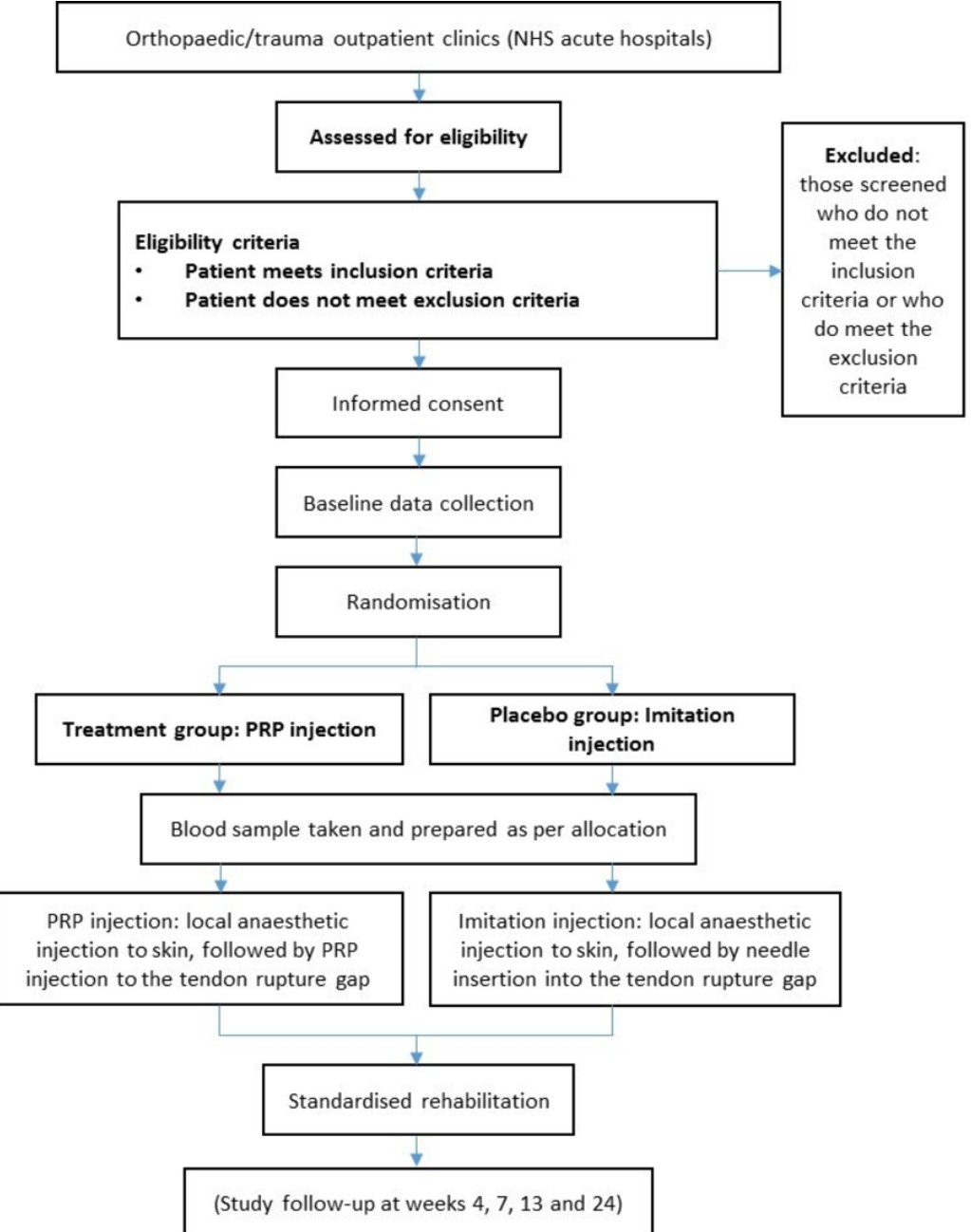

**Figure 2** The patient pathway for the main study. The substudies are not shown in this diagram. NHS, National Health Service; PRP, platelet-rich plasma.

is kept for analysis. In the imitation group, the same size needle is inserted into the tendon gap, held briefly and withdrawn without injecting anything.

As this study requires a single treatment, adherence to the protocol consists of the participant receiving the allocated treatment. This will be monitored, and every instance of the participant not receiving the allocated treatment will be investigated. Unless they request to withdraw, these participants will be retained in the trial, to avoid missing data and for follow-up. However, if a participant is unable to receive a PRP injection for any technical reasons, they will receive an imitation injection, and this will be recorded.

Training in delivery of the PRP injection and the imitation injection will be provided by the trial team, and a step-by-step manual is provided to each site. Video training materials are available to site staff via an access-controlled website[25] but not supplied here because of the potential for non-blinding.

## BLINDING PROCEDURE
It will not be possible to blind the research nurse, surgeon or extended scope physiotherapist involved in treatment preparation or delivery due to the nature of the intervention. However, the participant and the assessor for the

---

**Box  Serious adverse events**

**An adverse event is considered a serious adverse event if it satisfies at least one of the following criteria:**
► Results in death
► Is life-threatening
► Requires in-patient hospitalisation or prolongation of existing hospitalisation
► Results in persistent or significant disability/incapacity
► Results in congenital anomaly or birth defect
► Another medical event judged important in the opinion of the investigator

---

primary outcome are blinded. The participant will be shielded from the injection preparation as this is done in another room or while the participant is waiting outside the clinic. To reduce the risk of non-blinding, participants in both intervention groups will wait for approximately 17 min, the duration of a cycle for the PRP preparation, and not be in direct sight of the machine. If the machine is not out of earshot, a dummy cycle will be run on the machine. The participant will receive the injection while lying face down and visually obscured from the procedure. The primary outcome assessor will not be aware of allocation when they perform the HRET at 24 weeks after treatment.

### Standardisation of other treatments

All participants receive standard care at the participating site. Standard treatment for the non-surgical population is usually immobilisation of the ankle in a cast, splint or boot during the initial clinic visit. Immediately after the intervention (PRP or imitation), the ankle will be immobilised in a cast, splint or boot. Standardisation of key elements of rehabilitation is required for this trial to reduce the risk of efficacy interference from substantial variation in rehabilitation. The following milestones are standardised:
► duration of initial ankle immobilisation postintervention is at least 3 weeks,

► position of the foot and ankle in equinus during the initial immobilisation,
► referral to physiotherapy for rehabilitation,
► avoidance of rigid full-time immobilisation or non-weight-bearing for more than 6 weeks.

Standardisation will not be required for the ankle splinting method or device, when weight-bearing is commenced or the specific exercise prescription. We will standardise rehabilitation by providing guidance to surgeons and physiotherapists in written form. Monitoring adherence with these guidelines will be assessed by asking participants questions relating to progress with rehabilitation during follow-up.

### Objectives and outcome measures

The primary objective is to evaluate the clinical efficacy of PRP in acute ATR in terms of muscle–tendon function. The primary outcome measure is work in Joules derived from the HRET, which is a validated objective measure of Achilles tendon–muscle unit function at 24 weeks to evaluate efficacy.[26] Preceded with a warm-up activity of walking for 5 min, the test is first performed on the uninjured limb and then on the injured side. The HRET involves the participant standing on one leg on a 10° incline box, raising the heel for maximum height and lowering the heel repeatedly at a rate of 30 raises per minute guided by a digital metronome until task failure. Vertical displacement data from the movements are obtained using a computer-controlled linear encoder (Encoder, MUSCLELAB; Ergotest Innovation AS, Porsgrunn, Norway) that measures the height of each heel-rise repetition and integrates the data into custom-made software (PATH-2, MUSCLELAB; Ergotest Innovation AS, Porsgrunn, Norway). Work (Joule) is calculated as the product of body mass (kilogram, measured on class III scales), total vertical displacement (metre) and the constant 9.807 converting kilopond-metres to Joules. We will also report the number of repetitions and maximum heel-rise height (centimetre). The performance of the injured limb for each of the three variables will be

---

**Table 4  AEs not qualifying as SAEs**

| Foreseeable AEs | Unforeseeable AEs |
|---|---|
| May be reported if related to study treatment<br>► Bruising and discomfort at the venesection site<br>► Mild discomfort or minor bleeding from ATR site following injection<br>► Syncopal (fainting) episode associated with venesection or tendon injection<br>► Discomfort at ATR site during rehabilitation<br>► Technical complications of the lower leg casting and splinting<br>► Consequences of depending on walking aids<br>► Swelling or bruising of the lower leg and foot<br>► Deep vein thrombosis in a lower limb<br>► Rerupture of the treated Achilles tendon | Will be reported if related to treatment.<br>For example:<br>► Serious infection of ATR injection site<br>► Skin breakdown or ulceration of treated lower leg other than 'plaster sores'<br>► Severe pain requiring more than simple analgesia beyond 10 days after injection |

AE, adverse event; ATR, Achilles tendon rupture; SAE, serious adverse event.

**Table 5**  Protocol substantial amendments

| Amendment/date | Nature of amendment | Rational |
|---|---|---|
| SA01<br>3 June 2015 | ▶ Record maximum height and number of heel rises in HRET | ▶ Provide additional validation of the outcome measure |
| | ▶ Ask patient which intervention they think they received in 24-week postassessment questionnaire | ▶ Assessment of success of blinding strategy |
| | ▶ Stipulate guidelines for rehabilitation | ▶ To accommodate local preferences while ensuring the integrity and success of the trial |
| | ▶ Added guidance if allocated intervention cannot be given | ▶ Guidance was omitted in original protocol version |
| | ▶ Clarifications on the nature of the injury | ▶ Clarify injury type |
| SA02<br>8 March 2016 | ▶ Change inclusion criteria | ▶ Increase upper age limit with requirement of ambulatory status |
| | ▶ Increase recruitment period | ▶ 12 days postinjury instead of 7 |
| | ▶ Extended scope physiotherapists can administer the intervention | ▶ Pragmatic approach to accommodate for clinical practice |
| | ▶ Clarification of the ATR diagnosis | ▶ Clarification of the rupture location |
| | ▶ Clarification of anticoagulation | ▶ VTE prophylaxis requirement |
| | ▶ Randomisation and statistical alterations | ▶ Approval of randomisation and statistical plan |
| SA03<br>21 April 2017 | ▶ Extended 24 months follow-up | ▶ To study PRP on effect on the quality of the repaired Achilles tendon at 2 years postinjury |
| SA04<br>24 July 2017 | ▶ Extend recruitment by 2 months<br>▶ Increase sample size to 230 | ▶ DSMC blinded interim data analysis found HRET SD is 24 with larger variability in data. Sample size increased to guarantee 80% power |

ATR, Achilles tendon rupture; DSMC, Data and Safety Monitoring Committee; HRET, heel-rise endurance test; PRP, platelet-rich plasma; VTE, venous thromboembolism.

expressed relative to the uninjured limb by computing a Limb Symmetry Index (LSI)=(/)×100. Participants are asked to stop by the assessor if any of the following is observed: volitional stopping due to fatigue, inability to keep pace with the metronome or maintain full knee extension of the standing leg or using more than fingertip support against the wall. The assessor uses verbal prompts if the above are observed and stops the test if the participant does not respond to two consecutive prompts. The test will be standardised through providing study-specific encoder and software, HRET training manual and one-to-one training to the blinded assessor in each centre. Since the encoder is a very sensitive device, it records even minimal movements that might not correspond to actual heel rises. To dismiss potential measurement errors, two researchers (at least one being a physiotherapist) blinded to treatment allocation will independently review the videos of all assessments and discount any invalid repetition included in the HRET data. The primary and secondary outcome measures are presented in tables 2 and 3.

In the two substudies, the outcome measures are defined according to the substudy analysis. The whole blood and PRP component substudy includes analysing full blood cell count (red cells, platelets and white cells) in all participants in both groups and analysing PRP components in the PRP group. PRP analysis includes: cell count, platelet activation status (basal resting status and after ex vivo activation) as a measure of quality and relevant growth factor concentrations (PDGF, IGF-1, VEGF, fibroblast growth factor and TGF-β). All blood and PRP samples will be anonymised, placed in secure biological sample transport packaging and sent via tracked courier to a central laboratory (Institute of Inflammation and Ageing Laboratory at the University of Birmingham) for analysis.

In the immunohistochemistry substudy: 16 participants in one centre (Oxford) who have given informed consent to undergo the sample collection procedure will have tendon needle biopsy under ultrasound guidance at week 6 postintervention. The patients for this substudy will be the first eight consented in each arm at the Oxford site only, and the procedure is carried out by an experienced radiologist. Analysis includes tissue morphology, proliferation, apoptosis, vascularity, metabolic indicators and stem cell marker. The tissues will be stored for future analysis of relevant markers.

Data will be collected for all participants including those who do not receive the allocated intervention, unless they have withdrawn consent for follow-up. In order to obtain

as complete set of outcome data as possible, several attempts will be made to contact participants by phone for follow-up at each time point, and if they are not available, the questionnaire will be sent by post.

## Study flow chart

The patient pathway and main study activities are shown in figure 2.

## Safety reporting

Since PRP is prepared from autologous blood, concerns of disease transmission and immunogenic reactions are eliminated. Although there have been no serious adverse events (SAEs) related to using PRP reported in the literature, we have systems in place to monitor all adverse events (AEs) and SAEs. AEs will be collected during the study treatment episode, and staff should report any events they become aware of up to and including the 24 months follow-up appointment. SAEs must be reported to the Chief Investigator within 24 hours of the local research team becoming aware of the event. Participants will be asked if they have experienced any complications during follow-up data collection (box, table 4). Non-blinding of participants is not anticipated unless there are compelling medical or safety reasons. If non-blinding is requested by a site PI, the CI and OCTRU will make a decision based on the reasons for the request. Deviations from protocol and other unexpected events will be recorded on an incident form and assessed by the study team for implications for the study.

## Data management

All data will be processed according to the Data Protection Act 1998, and all documents will be stored safely in confidential conditions. Data will be collected from participants and site personnel via paper case report forms which will be returned to the central trial office by post using a Freepost address (prepaid). Blood samples and needle biopsy samples sent for analysis will be anonymised at source and only identified using the unique study number and participant initials. The HRET test data are transferred via the linear encoder linked to a study-dedicated laptop then transferred from each site to Oxford by an encrypted Universal Serial Bus following each participant HRET, and the original data remain on the site laptop. Blood and PRP samples will be stored at the Centre for Translational Inflammation Research at the Birmingham University Research Laboratory and disposed of at the end of the study. Needle biopsy samples for those participants taking part in the substudy (n=16) will be stored in the Oxford Musculoskeletal Biobank. Data provided from the blood sample analysis or biopsy samples analysis will be entered into the study database in Oxford. All data transfers will use appropriate password protected and/or encrypted files. The study management team will conduct data entry into a study-dedicated database which is developed and maintained by the OCTRU.

## Sample size and statistical analysis

Two hundred and thirty patients (115 in each arm) will provide 80% power to detect a standardised difference of 0.4 in the LSI from the HRET measure of work at 24 weeks postrandomisation and with 5% (two-sided) significance allowing for 20% loss to follow-up. This is based on previous data[18] where a clinically important difference of 10% with SD of 20% was observed and blinded interim data analysis that showed SD of 24% for the overall sample. This sample size will also provide 90% power and 5% (two-sided) significance to detect an effect size of 0.5 in the ATRS, based on a difference of 11 and SD of 21.4 (18). All comparative results will be presented as summary statistics with 95% CIs and reported in accordance with the non-pharmacological extension to the Consolidated Standards of Reporting Trial (CONSORT) statement.[27 28]

The primary statistical analysis will be carried out on the basis of intention-to-treat. The time window for collecting the HRET will be 2 weeks before the 24 weeks time point and up to 8 weeks after. The primary analysis to investigate the adjusted effect of PRP on ATR recovery will be multivariate linear regression, using the LSI as dependent variable, treatment as the main independent variable and the stratification factors plus other prognostic factors as additional independent variables. If the primary outcome data are normally distributed, the two groups (PRP×imitation injection) will also be compared using an unpaired Student's t-test to explore the unadjusted effect of the intervention. If data on the primary outcome are not normally distributed, the first approach will be data transformation, but if normality cannot be achieved, a non-parametric statistical test without adjustment will be used.

The PROMs (secondary outcomes) will be analysed in a linear mixed-effects model with longitudinal framework to allow data collected at all time points to be taken into account. This is a robust procedure that deals with some missing values; however, missing data imputation will be carried out if necessary. Similarly to the primary outcome, unadjusted analysis and data transformation (if necessary) will also be performed for all continuous secondary outcomes. Descriptive statistics (such as means and SD for continuous variables and frequencies and proportions for categorical or binary variables) will be used to describe the baseline characteristics of the participants in the two study groups; however, no formal statistical tests will be used to compare the groups.

For the two substudies, statistical analyses will be primarily descriptive, and the relationship between various biomarkers and clinical outcomes will be explored.

A separate statistical analysis plan (SAP) will contain full details of all statistical analyses and will be prepared early in the trial, agreed with the Data and Safety Monitoring Committee (DSMC) and finalised prior to the primary analysis database lock and before non-blinding of the data. Any changes at this time will be incorporated into the final SAP and signed off as per current OCTRU

standard operating procedures (SOPs). Any changes/deviations from the original SAP will be described and justified in the protocol and/or in the final report, as appropriate. Comparative outcome interim analyses are not planned unless requested by the DSMC.

## Ethics and dissemination

This trial will be conducted in accordance with the ethical principles that have their origin in the Declaration of Helsinki and that are consistent with GCP and the applicable requirements as stated in the Research Governance Framework for Health and Social Care (second edition 2005). The study may be monitored or audited in accordance with the current approved protocol, GCP, relevant regulations and SOPs.[29]. An independent DSMC is established to safeguard the interests of trial participants, potential participants and future patients, to assess the safety and efficacy of the interventions during the trial and to monitor the overall conduct of the trial, protecting its validity and credibility. DSMC will adopt the DAMOCLES charter[30] and meet at least annually, with an option to increase if specific concerns arise. Local NHS trust approval and contract with the sponsor is required before recruitment initiation at each site. The study may be audited by the sponsor or the clinical trials unit. The study office team will conduct monitoring visits to sites as defined in the Risk Assessment and Monitoring Plan.

The trial will be reported in accordance with the CONSORT statement[31] and its extensions relating to non-pharmacological studies[27 32] and TIDieR (template for intervention description and replication) guidelines for intervention description and replication.[33]. A summary of the trial outcome will be disseminated to trial participants on relevant websites and by email, where an email address is provided. In addition to the NIHR monograph report, the results will be published in peer-reviewed medical literature and may be presented at relevant national and international conferences.

## Amendments to the protocol

Amendments will be handled through Health Research Authority (HRA) procedures. Two substantial amendments have been implemented since the study began (table 5). The first amendment was submitted prior to commencement of the trial to add additional information, rehabilitation guidance and blinding assessment. The second amendment was performed after 40 patients were recruited to change recruitment criteria, clarify diagnosis, clarify anticoagulation and update the randomisation mechanism and statistical plan. The change to randomisation was required due to imbalance in participants' age group stratum following a systems issue. The underlying systems issue was fixed, and a change to the randomisation strategy was implemented to avoid this imbalance being preserved throughout the study. The randomisations allocated prior to the change were not altered. This approach was reviewed and approved by the sponsor, DSMC, Trial Steering Committee (TSC) and the Ethics Committee.

## DISCUSSION

This multicentre trial opened to recruitment in July 2015 and will reach recruitment targets in September 2017. The trial is due to report results in July 2018.

### Author affiliations
[1]Institute of Translational Medicine, University of Liverpool, Liverpool, UK
[2]Nuffield Department of Orthopaedics, Rheumatology and Musculoskeletal Sciences, Kadoorie Centre for Critical Care Research, John Radcliffe Hospital, University of Oxford, Oxford, UK
[3]Nuffield Department of Orthopaedics, Rheumatology and Musculoskeletal Sciences, University of Oxford, Oxford, UK
[4]Institute of Inflammation and Ageing (IIA), University of Birmingham Laboratories, Birmingham, UK
[5]Faculty of Health and Human Sciences, School of Health Professions, University of Plymouth, Plymouth, UK
[6]Oxford Clinical Trials Research Unit, Nuffield Department of Orthopaedics, Rheumatology and Musculoskeletal Sciences, Centre for Statistics in Medicine, University of Oxford, Oxford, UK

**Acknowledgements** This trial is being performed on behalf of the PATH-2 study group under the Trial Steering Committee whose independent members are: Professor Roger Smith (Chair), Dr Catriona Graham, Dr Dylan Morrissey, Professor John O'Byrne, Dr Gustaaf Reurink and Mrs Sarah Webb and the Data and Safety Monitoring Committee: Dr Chao Huang (Chair), Mr Andy Goldberg and Dr Carey McClellan. The trial team would like to thank the following principal investigators (PIs) and their teams for their invaluable contribution at each site: Robert C Handley, Maneesh Bhatia, Andrew Kelly, Steve Hepple, Michael Carmont, Paul Hodgson, Nima Heidari, Jonathan Young, Gareth Stables, Ravindran Ranjith, Simon Frostick, Jim Carmichael, Claire Topliss, Lyndon Mason, Nasser Kurdy, Mark Davies, Adrian Hughes, Simon Barnes and Matthew Solan.

**Contributors** JA developed the protocol, contributed to the writing of the paper and conceptual work for the trial, is a member of the trial management group (TMG) and is a grant coapplicant. DJK is the research fellow for the trial, developed the study protocol, contributed to the writing of the paper and design of the primary outcome assessment, is a member of the TMG and a grant coapplicant. PAH developed the protocol, is a member of the TMG and is a grant coapplicant. PH developed the protocol, is a member of the TMG and is a grant coapplicant. SW is the current trial manager and contributed to the writing of the paper .CB is the research physiotherapist for the trial and contributed to the design of the primary outcome assessment. MMS is the current trial statistician for the study, developed the statistical analysis plan and contributed to the writing of the paper. SJD developed the protocol and statistical aspects of the study, is a member of the TMG and is a grant coapplicant. SEL developed the protocol, is a member of the TMG and is a grant coapplicant. KW is the chief investigator, contributed to the writing of the paper and is the senior grant holder. All authors contributed to the refinement of the study protocol and approved the final manuscript.

**Funding** This project is funded by the Efficacy and Mechanism Evaluation (EME) Programme, a Medical Research Council (MRC) and NIHR partnership (reference no 12/206/30). The sponsor for the PATH-2 study is the University of Oxford. Monitoring the conduct of the study is delegated to the Clinical Trials and Research Governance team.

**Disclaimer** The views expressed in this publication are those of the author(s) and not necessarily those of the MRC, NHS, NIHR or the Department of Health.

**Competing interests** None declared.

**Ethics approval** Oxfordshire Research Ethics Committee (reference no 14/SC/1333).

**Provenance and peer review** Not commissioned; externally peer reviewed.

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
