## [Reviewer comments · BMJ Open]

ARTICLE DETAILS

TITLE (PROVISIONAL)	Platelet Rich Plasma in Achilles Tendon Healing (PATH-2) Trial: Protocol for a multi-centre, participant and assessor-blinded, parallel-group randomised clinical trial comparing Platelet Rich Plasma (PRP) injection versus placebo injection for Achilles tendon rupture
AUTHORS	Alsousou, Joseph ; Keene, David; Hulley, Philippa; Harrison, Paul; Wagland, Susan; Byrne, Christopher; Maia Schlüssel, Michael; Dutton, Susan; Lamb, Sarah; Willett, Keith

VERSION 1 REVIEW

REVIEWER	MEHMET HASAN TATARİ Dokuz Eylül University Medical Faculty Dep. of Orthopedics and Traumatology, İzmir, Turkey
REVIEW RETURNED	13-Aug-2017
GENERAL COMMENTS	This is a very well designed and detailed study with all aspects. Congratulations and have a good lock.

VERSION 1 AUTHOR RESPONSE

Reviewer: 1

Reviewer Name: MEHMET HASAN TATARİ

Institution and Country: Dokuz Eylül University Medical Faculty Dep. of Orthopedics and Traumatology, İzmir, Turkey

Please state any competing interests: None declared

Please leave your comments for the authors below

Comment: This is a very well designed and detailed study with all aspects. Congratulations and have a good lock.

Response: We are constantly trying to find ways of improving our peer review system and continually monitor processes and methods by including article submissions and reviewers' reports in our research. If you do not wish your paper or review entered into a our peer review research programme, please let us know by emailing The BMJ's editorial office papersadmin@bmj.com as soon as possible.